# Distant sequence regions of JBP1 contribute to J-DNA binding

Ida de Vries[1] , Danique Ammerlaan[1], Tatjana Heidebrecht[1] , Patrick HN Celie[1] , Daan P Geerke[2] ,
Robbie P Joosten[1] , Anastassis Perrakis[1]

**Base-J (β-D-glucopyranosyloxymethyluracil) is a modified DNA nucleotide that replaces 1% of thymine in kinetoplastid flagellates. The biosynthesis and maintenance of base-J depends on the base-J-binding protein 1 (JBP1) that has a thymidine hydroxylase domain and a J-DNA-binding domain (JDBD). How the thymidine hydroxylase domain synergizes with the JDBD to hydroxylate thymine in specific genomic sites, maintaining base-J during semi-conservative DNA replication, remains unclear. Here, we present a crystal structure of the JDBD including a previously disordered DNA-contacting loop and use it as starting point for molecular dynamics simulations and computational docking studies to propose recognition models for JDBD binding to J-DNA. These models guided mutagenesis experiments, providing additional data for docking, which reveals a binding mode for JDBD onto J-DNA. This model, together with the crystallographic structure of the TET2 JBP1-homologue in complex with DNA and the AlphaFold model of full-length JBP1, allowed us to hypothesize that the flexible JBP1 N-terminus contributes to DNA-binding, which we confirmed experimentally. A high-resolution JBP1:J-DNA complex, which must involve conformational changes, would however need to be determined experimentally to further understand this unique underlying molecular mechanism that ensures replication of epigenetic information.**

## Introduction

The modified nucleotide β-D-glucopyranosyloxymethyluracil (base-J) replaces 1% of the thymine nucleotides in kinetoplastid flagellates (1). Base-J is found mainly in telomeric repeats and other repetitive sequences. In *Leishmania*, 99% of base-J is found in telomers, mainly in GGGTTA repeats, wherein the second thymine is modified to base-J (2, 3, 4). The remaining 1% of base-J is involved in transcription. In *Leishmania*, base-J has been shown to be involved in transcription

termination (5, 6), whereas in *Trypanosoma* base-J is marking transcription initiation (7). The biosynthesis of base-J occurs in two steps (Fig 1A). In the first step, the methyl group of a thymine is hydroxylated by JBP1 or JBP2 (8, 9). Both have a β-oxoglutarate and $Fe^{2+}$ thymidine hydroxylase domain (THD), which is a functionally divergent homologue also present in human TET proteins, an important finding that established the TET-JBP family (10, 11, 12). In the second step, the base J-associated glucosyltransferase adds a sugar moiety to the hydroxymethyluracil intermediate, resulting in base-J (13, 14, 15). More recently, another protein bearing a J-DNA-binding domain (JDBD), JBP3, has been identified and shown to be involved in transcription termination in trypanosomes and *Leishmania* (16, 17).

*Leishmania* species are unicellular eukaryotic parasites, infecting humans and their livestock, resulting in disease and economic losses in many countries around the world (18). *Leishmania* parasites are transmitted by sand flies to the vertebrate host, in which they infect macrophages, where they grow and divide, and ultimately lysing the host cell and infecting other macrophages (19). The severity of the infection can vary, depending on the species and several host factors (20). The nuclear genomes of Leishmania species are organized into polycistronic transcription units containing tens to hundreds of protein-coding genes (21), and thus, *Leishmania* genomes contain few transcription start sites and transcription termination sites (TTSs): the non-telomeric 1% of base-J is located in TTSs. Loss of internal base-J leads to massive read through of transcriptional stops by RNA polymerase II, which is lethal for *Leishmania* (5, 6). This mechanism is not conserved in all kinetoplastids. For example, in *Trypanosoma brucei*, transcription termination occurs in a base-J-independent manner (6), and loss of base-J is not lethal. Thus, understanding the enzymes involved in J-DNA biosynthesis and replication is important to both characterize them as potential drug targets, but also to understand potential species differences that might be related to the essentiality of base-J for the life circle of these parasites.

JBP1 (Fig 1B) specifically recognizes base-J-containing double-stranded DNA (J-DNA) (22, 23) through a JDBD which has been

[1]Oncode Institute and Division of Biochemistry, Netherlands Cancer Institute, Amsterdam, The Netherlands   [2]Department of Chemistry and Pharmaceutical Sciences, Amsterdam Institute of Molecular and Life Sciences (AIMMS) and Amsterdam Center for Multiscale Modeling (ACMM), Vrije Universiteit Amsterdam, Amsterdam, The Netherlands

Correspondence: a.perrakis@nki.nl

 

**Figure 1. JBP1 is involved in the biosynthesis of base-J.**
**(A)** Biosynthesis of base-J (β-D-glucopyranosyloxymethyluracil). Thymine is hydroxylated by JBP1 or JBP2 to form hmU (hydroxymethyluracil), which is glucosylated by JGT to result in the modified nucleotide base-J. R indicates the rest of the nucleotide of thymine, which is located in a DNA strand. **(B)** Architecture of JBP1, which contains a JDBD domain which is in sequence inserted into the THD.

structurally characterized (24). In the context of JBP1, JDBD recognizes base-J and then probably invokes a conformational change in JBP1 (25, 26), enabling the THD to initiate the biosynthesis of base-J on a thymine 13 base pairs downstream the complementary DNA strand (J + 13′ position) (27). This synergy between recognition of a preexisting base-J in the parental strand, and hydroxylation of a thymidine in the daughter strand, is likely important for replication of the base-J epigenetic marker.

It has been previously shown that the C2 and C3 hydroxyl groups of the sugar moiety of base-J interact with the non-bridging phosphoryl oxygen of the nucleotide that is positioned before base-J (J-1 position), ensuring that the sugar moiety is positioned perpendicular towards the major groove of the J-DNA helix (28). That would leave the C4 and C6 hydroxyl groups free for recognition of J-DNA by the JDBD (24). Structure-based mutagenesis and functional experiments have shown Asp525 in the JDBD to be adequate in explaining the discrimination between J-DNA and standard DNA oligonucleotides. The D525A mutation results in ~1,000 times worse binding to J-DNA and 10 times better binding to regular DNA, abolishing all specificity. This strongly indicates that Asp525 forms hydrogen bonds with the free C4 and C6 hydroxyl groups of base-J, as both are found to be essential in forming the JBP1:J-DNA complex. Further mutation studies showed that residues Lys518, Lys522, and Arg532 are important for the affinity of JBP1 towards both J-DNA and regular DNA (24). This indicates that these residues are not involved in the specificity for J-DNA, but are important in formation of a JBP1:DNA complex. Combining these conclusions with the crystal structure of the binding domain (PDB-ID: 2XSE) and with results from hydrogen deuterium exchange rate (HDX-MS) analyses, a manually built model for the JDBD:J-DNA complex has been proposed and validated by small-angle X-ray scattering experiments (24). Notably, the crystal structure of the JDBD had a missing loop of eight residues (529–537) between helices α4 and α5, including the Arg532 that is important for DNA binding. This loop had been modeled by hand, but had not been observed in the electron density maps of the structure.

The crystallization and structure determination of JDBD was used as a demonstration project during the Cold Spring Harbor Laboratory course. JDBD is one of the proteins that were provided to the students, and they were asked to crystallize it, collect diffraction

data, and determine the structure. Remarkably, in one of numerous datasets that have been obtained during the course, the missing density for the loop 529–537 was observed. Motivated by this new structure in which we were able to build that missing loop, we performed molecular dynamics (MD) simulations in combination with docking studies and determined a new and informative structure model representative for the JDBD:J-DNA complex, which was further refined by mutagenesis experiments and validated with small-angle X-ray scattering (SAXS) data. The recent availability of AlphaFold structure prediction of the full-length JBP1 protein (29, 30) together with the X-ray structure of the TET2 homologue of the THD in complex with DNA (31) and the JBP1-JDBD:J-DNA model from this study allowed to further understand how JBP1 binds J-DNA and to demonstrate that the N-terminal helical region participates in this binding.

## Results

### A new crystal structure for JDBD with an ordered loop

The previously published crystal structure of JDBD (24) (PDB-ID: 2XSE) was missing residues 529–537. It has been noticed that the electron density maps from a dataset collected by students following the CSHL 2018 course on X-Ray Methods in Structural Biology, showed rather clear density for that loop. As this loop follows the α4 recognition helix and harbors the Arg532 residue which affects DNA binding (24), the diffraction data from the CSHL 2018 course were reprocessed, and the model was rebuilt and refined. The final structure was validated using MolProbity (32) and it belongs to the 100$^{th}$ percentile, with a Ramachandran Z-score (33) of −0.98 ± 0.72. The new structure at a resolution of 1.95 Å adopts a highly similar conformation as 2XSE, which had been refined at a nearly identical resolution (1.90 Å). The space group is identical (P6$_1$22) and the cell dimensions do not differ substantially (a, b, c: 67.34, 67.34, 186.76 Å for 2XSE versus. a, b, c: 68.29, 68.29, 185.84 Å for 8BBM). However, the new crystal structure contains all residues from 392 to 561 (Fig 2A, [omit map in Fig S1]), including the 529–537 loop which is well resolved (Fig 2B and C). The B factors in the loop range between 53.5 and 125.2 (mean 91.6) compared with a mean of

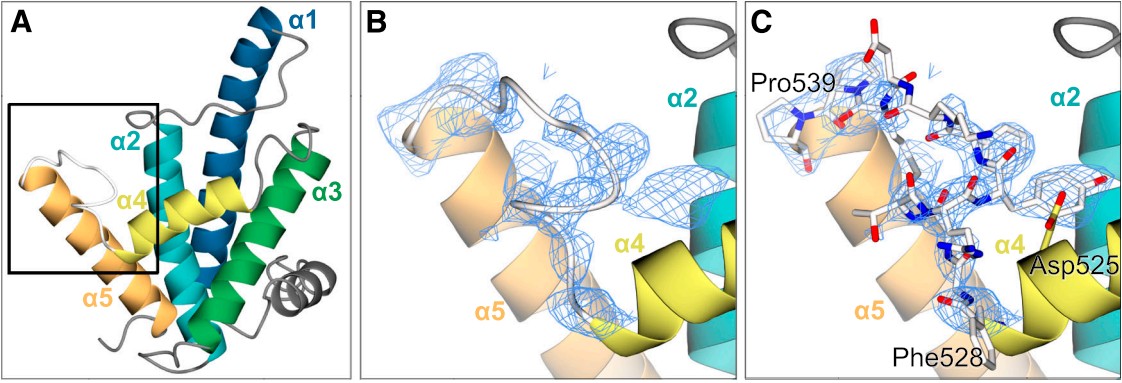

**Figure 2. Crystal structure of the J-DNA-binding domain with an ordered loop.**
**(A, B, C)** New crystal structure of JDBD (PDB-ID: 8BBM) with the black box indicating the zoom-in area of (B, C). **(B, C)** Electron density ($2mF_o$-$DF_c$ map contoured at 1 rms) observed for loop 528–539 in the new crystal structure (PDB-ID: 8BBM); the 528–539 loop is shown as ribbon (B) and with all atoms (C).

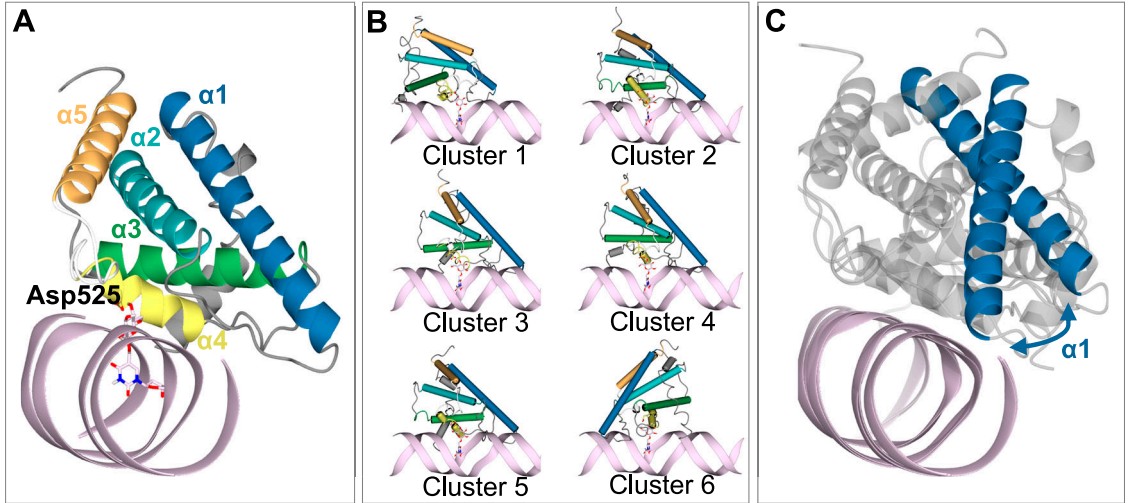

**Figure 3. Docking models of the J-DNA-binding domain (JDBD) in complex with J-DNA.**
**(A)** Docking output of docking J-DNA (pink) onto JDBD new crystal structure with base-J in pink and Asp525 in yellow cylinders. **(B)** Docking output using the center structures of the six most occupied clusters obtained from clustering MD simulations as the protein template. JDBD is shown as tubes; Clusters 1, 2, and 3 originate from MD simulation run 1, Clusters 4, 5, and 6 from MD simulation run 2. **(C)** The blue arrow is illustrating the flexible orientation of the α1 helix perpendicular to the J-DNA as observed in the JDBD:J-DNA complexes obtained from docking. The overlay is generated from the best scoring structure of the most occupied cluster obtained from docking with Cluster 1 and Cluster 2 as the protein template.

65.3 for the JDBD structure excluding the five α-helices. There are no crystal contacts observed for this loop, and there are no substantial differences in crystal properties overall. Thus, it remains unclear why the 529–537 loop is ordered in the new structure and not in previous experiments.

### Investigating the JDBD interaction with J-DNA by computational modeling

We then wanted to use our new crystal structure of JDBD for docking to a J-DNA model to investigate if that would lead to new insights in the possible binding mode of JDBD to J-DNA. Whereas previously we have done such studies manually, here we decided to follow a computational approach. In the manually created model (24), base-J is oriented perpendicular towards the major groove of the DNA and the interactions of the O2 and O3 hydroxyl groups of base-J with the

non-bridging phosphoryl oxygen at J-1 in J-DNA are modeled, as first observed by reference 28. To now be able to create a computational model, parameters to model base-J and J-DNA were generated and implemented in HADDOCK2.4 (34). In the docking protocol, JDBD was docked onto a 20-mer J-DNA helix. The Asp525 residue was restrained to be in the proximity of the C4 and C6 hydroxyl groups of the sugar moiety of base-J in both complexes, based on its importance for recognizing base-J.

The docking study using JDBD resulted in three HADDOCK clusters with docked structures, of which the most occupied cluster contains 93% of the generated models. The best scoring model is shown in Fig 3A. The JDBD crystal structure was then used to start MD simulations to explore the behavior and flexibility of this protein domain in solution and obtain additional templates for further docking studies. Two parallel simulations were run, in which both the Cα root-mean-square deviation (RMSD) and root-mean-square fluctuation values

Mechanism of J-base recognition by JBP1   de Vries et al.   https://doi.org/10.26508/lsa.202302150   vol 6 | no 9 | e202302150   **3 of 13**

were comparable (Fig S2A and B). The overall fold of the protein is maintained as seen by clustering the obtained structures (Fig S3), with, as expected, more flexibility for loops and termini compared with residues located in the helices (Fig S2B). As in both MD simulations, most obtained conformations are represented by the three most occupied clusters (93% coverage in simulation run1, 64% coverage in run2, Fig S4A–D); we assume that the center structures of JDBD represented in these six clusters provide a representative set of structures for the behavior and flexibility of JDBD in solution.

The six selected representative JDBD models from clustering the MD trajectories were used to investigate if protein flexibility would affect the JDBD:J-DNA model proposed by HADDOCK. In the obtained complexes, the overall positioning of JDBD along the J-DNA strand is consistent (Figs 3B and S5A–E and Table S1). The α1 helix is positioned towards the J-DNA, but has a different orientation in Cluster 1, compared with Clusters 2, 3, 4, and 5 (Fig 3C). The model obtained using Cluster 6 as the JDBD template shows a mirrored protein orientation with respect to the J-DNA compared with the models obtained using the centers of Clusters 1–5 as the protein template (Figs 3B and S5F). Although this mirrored orientation is preferred in most of the HADDOCK models (69%), the remaining 31% of the structure models obtained from the docking run with Cluster 6 is similar to the models in Clusters 1–5.

To validate the docking models, the obtained structures were compared with experimental SAXS and HDX-MS data presented in previous work (24). The experimental SAXS curves fit well to the calculated curve for the models obtained from docking, as indicated by the $\chi^2$ values (Fig S6A–F). Furthermore, the HDX-MS data were mapped onto the docking output (Fig S7A–F), demonstrating that the α4 helix (which contains the Asp525 residue) and the α1 helix (which is positioned towards J-DNA as in the docking models) show the most pronounced HDX-MS reduction.

The interaction of base-J with Asp525 and the known intra-molecular interaction with the phosphate group at J-1 that were used as explicit HADDOCK restraints, and are respected in the models (Fig S8A–E). Residues Lys518 and Lys522, which were previously suggested to be involved in complex formation (24), do not form direct interactions with J-DNA in the obtained models. On the other hand, in the complexes with JDBD Clusters 1, 2, and 4, Arg532 seemingly forms hydrogen bonds with the non-bridging phosphoryl oxygen of the nucleic acid two base pairs that are further along the opposite strand (J + 2′ position) (Fig S8A, B, and D). In Clusters 3 and 5, this arginine is positioned such that hydrogen bond formation with the non-bridging phosphoryl oxygen of the nucleic acid base pair at position J + 3′ can be possible (Fig S8C and E). However, the distance between the NH hydrogen of Arg532 and the non-bridging phosphoryl oxygen in the DNA is too long (≥2.5 Å) to form direct hydrogen bonds in all models. Because a docking model is a static representation of the JDBD:J-DNA complex, it is likely that hydrogen bonds or salt bridges are formed with the phosphoryl group at the J + 2′ and/or the J + 3′ position to stabilize the negative charges in the J-DNA backbone.

## Computational modeling provides new insight into JDBD mediated J-DNA recognition

Having verified that our previous hypotheses about the JDBD interaction with J-DNA were basically correct, we examined the

models for novel insights focusing on charge-conserved residues in the α1–α2 region that are interacting with J-DNA. Based on multiple-sequence alignment of JBP1 in 28 different species (Fig S9), Glu437, His440, Arg448, and Asn455 were selected for site-directed mutagenesis. Arg448 and Asn455 are located in the α1–α2 loop, fully conserved and in proximity to the J-DNA when the α1 helix is oriented towards the DNA. Both residues are also located within the part that shows HDX-MS reduction (9% for peptide 442–463) (Fig 4A and B). The negative charge of residue 437 is conserved in 93% of the species and 86% of residues present at position 440 are positively charged (Fig S8). Both Glu437 and His440 are pointing towards the DNA and are also positioned within the region that shows high HDX-MS reduction (23% for the peptide 434–441) (Fig 4A).

The single-point mutants JDBD-E437A, -H440A, -R448A, and -N455A were produced and purified (Fig S10). For mutant E437A, the expression was substantially reduced compared with wt-JDBD and the other JDBD mutants, suggesting that this mutation could also affect the overall stability of the protein which could hamper binding to DNA. To address this issue, the thermal stability of wt- and mutant JDBD proteins was analyzed at two protein concentrations using nanoDSF. The E437A mutant showed about 9–12°C reduction in melting temperature compared with that of wt-JDBD and the other mutants (Table S2). The other mutants are as stable or slightly more stable compared with the WT, suggesting that the DNA-binding analysis is not affected significantly by altered protein stability. With all mutant proteins in hand, binding to both J-DNA and T-DNA was measured by fluorescence polarization. The wt-JDBD showed an affinity of 17 nM. Substitution of E437 and H440 only resulted in a minor difference in affinity for J-DNA (30 and 57 nM, respectively; Fig 4C and Table S3), indicating that these residues have a negligible contribution to the stabilization of the JDBD:J-DNA complex. In contrast, R448A showed a ~500-fold reduction in affinity to J-DNA (~8 μM) and N455A a ~400-fold reduction (~7 μM). The affinities of all four mutants and wt-JDBD towards T-DNA were all similar (>100 μM, Fig S11 and Table S3), indicating that mutation of Arg448 and Asn455 residues induces a loss in discrimination towards J-DNA (24).

Based on these new experimental insights, residues Arg448 and Asn455 were included as additional active residues in HADDOCK. The results show one clearly most occupied cluster that contains 79% of the generated structure models (Table S4). The best scoring structure in this cluster (cluster1_1.pdb) (Fig 5A and B) fits the SAXS curve obtained previously (24) (Fig 5C). In addition to the interactions found in our initial docking experiments, Arg532 forms hydrogen bonds with the non-bridging phosphoryl oxygens at position J + 2′. Interestingly, Asn455 forms a hydrogen bond with Lys522, which previously was found to be an important residue for the JDBD:J-DNA complex formation (24). This interaction supports Asn455 in forming an additional hydrogen bond with the J-DNA backbone to stabilize the negative charge. The residues Lys518 and Arg447 are also contributing in stabilizing the negative DNA charge distribution by orienting towards the DNA backbone and forming salt bridges with phosphate oxygen atoms. Such interactions are also observed for Arg448 and Arg517, of which, the latter was previously suggested to be relevant for complex formation (24). The residues described in the JDBD:J-DNA interface are in general highly

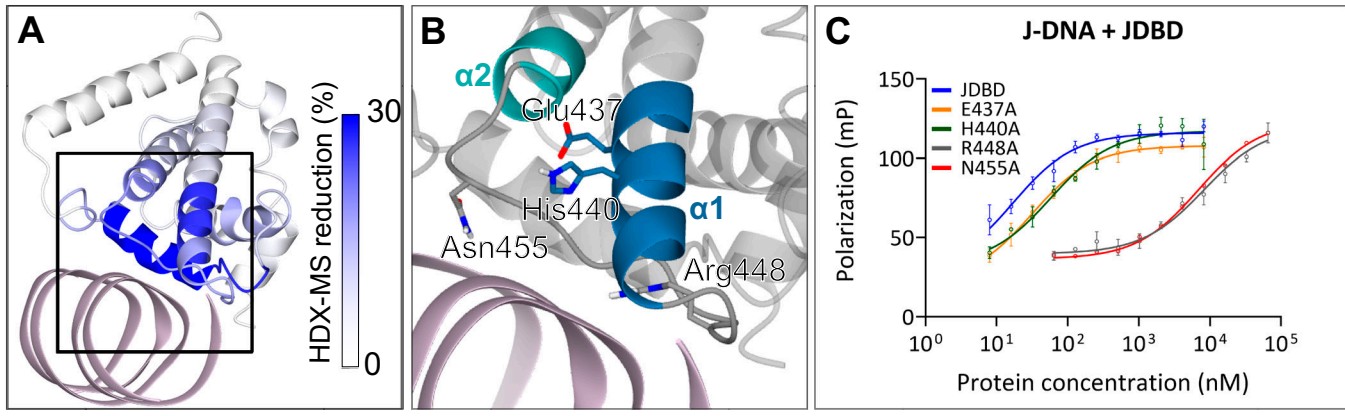

**Figure 4. Arg448 and Asn455, located in the α1 helix of J-DNA-binding domain (JDBD), are important for J-DNA recognition.**
**(A, B)** JDBD:J-DNA complex Cluster 2 colored by HDX-MS reduction that was mapped onto the complex. J-DNA is colored pink and the black box is indicating the zoom-in area of (B). **(B)** Zoom-in on cluster 2 indicating residues 434–463 of the protein with nontransparent colors and residues that were selected for mutagenesis studies indicated. The rest of the protein is colored in transparent grey and J-DNA in pink. **(C)** Polarization curves for wt-JDBD and single-point mutants with J-DNA.

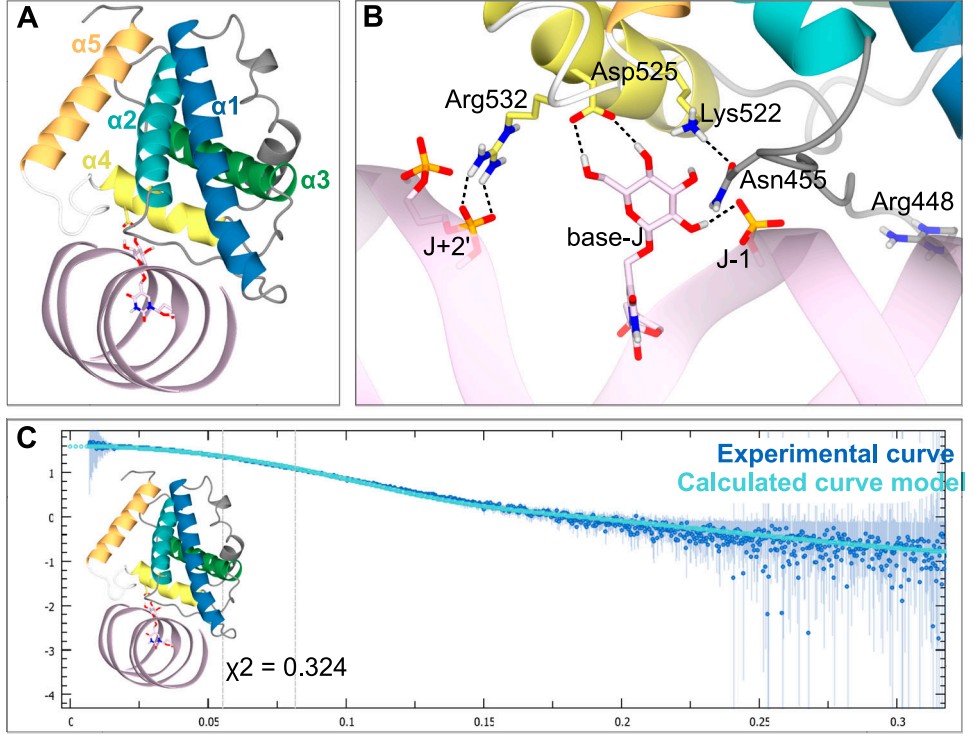

**Figure 5. A docking model of the JDBD:J-DNA complex incorporating all available data.**
**(A)** Overall docking model; J-DNA is colored in pink; base-J is shown as pink cylinders, and Asp525 is highlighted as yellow cylinders. **(B)** Binding site of this model with residues Lys522, Aps525, and Arg532 as yellow cylinders, Asn455 is highlighted as grey cylinders. Base-J is shown as pink cylinders and hydrogen bonds are indicated as black dotted lines. For clarity residues, 534 till 537 are not shown (in the white loop). **(C)** Experimental SAXS curve of JDBD:J-DNA (blue) compared with the calculated curve of the obtained docking model (cyan).

conserved (Fig S12). Asp525 is fully conserved in *Leishmania, Trypanosoma, Leptomonas*, and *Bodo saltans* species, so are Lys522, Arg532, and ArgR448. Asn455, which we identify in this study, is not conserved in *Leptomonas*, and Arg517 is not conserved in *Trypanosoma* also.

## New insights through AI-based modeling of JBP1

The AlphaFold structure prediction model (30) of JBP1 (entry Q9U6M1) (Fig 6A) is a source of new information that became available during the course of this study. In this predicted model,

JDBD shows a similar overall fold when compared with our new crystal structure (Fig 6B), with a noticeable difference in the loop between the α4 and α5 helices, that we found ordered in the new JDBD crystal structure. The THD of the AlphaFold model was compared with the hydroxylase domain of the human TET2 homologue (Fig 6C and D), which looks similar based on secondary structure elements (RMS difference of 0.35 over 120 selected Cα atoms). The AlphaFold model also shows that β-sheets are continuing between the N- and the C-terminal parts of the THD (Fig 6D) This observation confirms our previous results describing the THD domain (26), which is formed between the N- and C-terminal parts

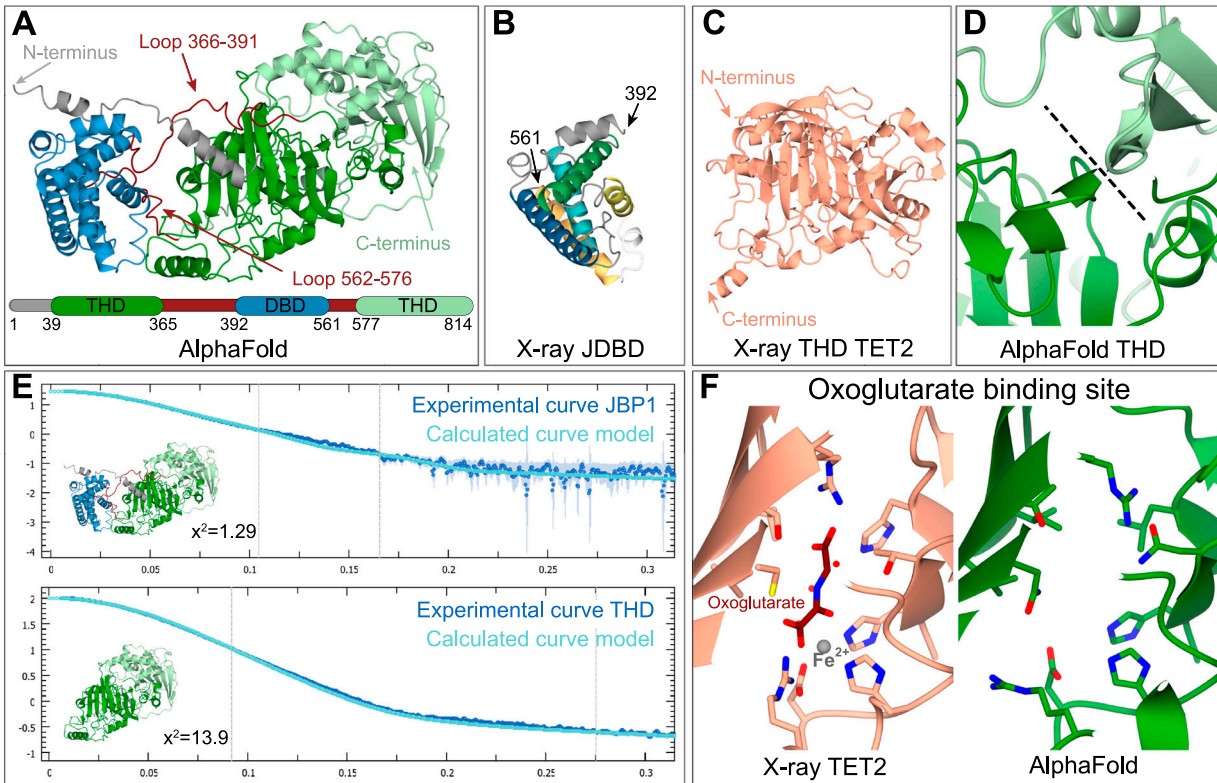

**Figure 6. The J-DNA-binding domain (JDBD) and thymidine hydroxylase domain (THD) in the AlphaFold model of JBP1 in *Leishmania tarentolae* are accurately predicted based on comparison with experimental data.**
**(A)** Structure of the AlphaFold model of JBP1, the different domains, and flexible regions are indicated by coloring and arrows. **(B)** X-ray structure of the JDBD as obtained in this work has the same fold as the JDBD in the AlphaFold model. **(C)** The THD of the JBP1 AlphaFold model adopts a similar fold compared with TET2 homologue. **(D)** The β-sheets between the N- and C-terminus of the THD are contiguous. The black dotted line indicates the interface. **(E)** Both the JPB1 model and the THD only are in agreement with the SAXS data obtained previously. **(F)** The oxoglutarate-binding site in the THD of the AlphaFold model is predicted properly compared with the homologues structure of TET2.

of JBP1 and retained enzymatic activity in vitro. The AlphaFold model fits the SAXS curve that was obtained previously (26) for the full-length JBP1 protein and the THD only (Fig 6E). The $\chi^2$ value of the fit is slightly elevated because of the presence of the connecting loops between the THD and the JDBD and the N-terminus in the protein used for measuring the SAXS curve, which were removed from the computational model. Examining the hydroxylation site in the THD domain of the AlphaFold model, the residues involved in $Fe^{2+}$ and oxoglutarate binding are in similar conformations as in the corresponding binding domain TET2 homologue (Fig 6F). These observations suggest that the AlphaFold prediction provides a reliable description of the JDBD and THD that can be used for further investigation.

The predicted aligned error (PAE) plot of the AlphaFold JBP1 model is consistent with the flexibility between the THD and the JDBD (Fig 7A), which we previously suggested based on experimental data (26). Interestingly, the PAEs for the N-terminal ~40 residues of JBP1 suggest that the N-terminal residues are reliably modeled in proximity to the JDBD. This PAE pattern stands for *Leishmania, Crithidia,* and *Leptomonas* species but not for *Trypanosoma* (Figs 7B and S13). The AlphaFold model for *Leishmania tarentolae* predicts two short helices in the N-terminal region, connected by a linker. The per-residue confidence scores (pLDDT)

range between "low" and "confident" (Fig 7C). In addition, the N-terminal region has positively charged residues that could, in principle, mediate DNA binding (Fig 7D). Finally, we have previously shown that the JDBD binds to J-DNA with an affinity about three times less than full-length JBP1 (24). These observations and previous findings prompted us to investigate if the JBP1 N-terminus plays a role in DNA recognition.

To this end, we decided to study the J-DNA binding affinity of JBP1 truncated after the first predicted N-terminal helix (at position 23, Δ23-JBP1) and after both N-terminal helices (at position 38, Δ38-JBP1). The truncated mutants were expressed and purified (Fig S10), and their affinity for J-DNA was studied by fluorescence polarization (Fig 7E). Whereas wt-JBP1 binds DNA with an affinity of 6.4 nM, Δ23-JBP1 binds J-DNA three times weaker (20.8 nM). Removal of all 38 N-terminal residues results in a further decrease in affinity to 31.2 nM (Table S5). These findings suggest that the N-terminal residues are involved in J-DNA binding. Notably, the affinity of the JDBD alone on J-DNA is 20.7 nM, which is similar to the N-terminally truncated JBP1. The derived free energy of binding (ΔG) of the JDBD alone or Δ23-JBP1 is about −48 kJ mol⁻¹, whereas the ΔG for wt-JBP1 is −45 kJ mol⁻¹. Thus, the difference in binding free energies between JDBD and the truncated mutants is relatively small (3 kJ mol⁻¹ or less). Based on all these data, we attempted to create a molecular model of JBP1 bound

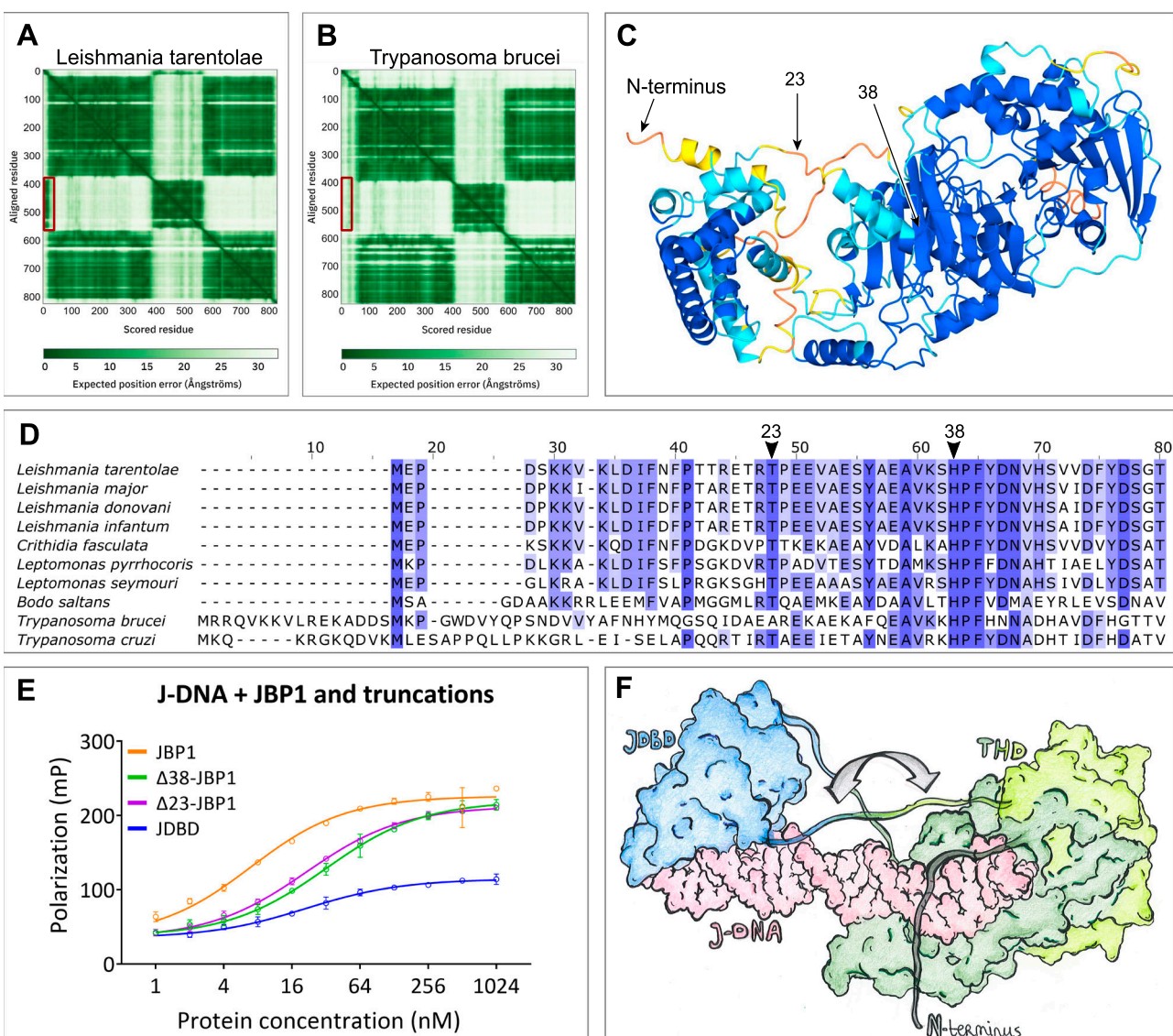

**Figure 7. An N-terminal sequence of JBP1 contributes to DNA binding.**
**(A, B)** The PAE plot for the model of JBP1 from (A) *L. Tarentolae* and (B) *Trypanosoma brucei* from the AlphaFold database (entries Q9U6M1 and P86937, respectively); the PAE regions indicative of the association of the N-terminal sequence with the JDBD are marked with a red box. **(C)** The AlphaFold model of the full-length JBP1 protein colored by pLDDT values; orange and yellow indicate very low- and low-confidence regions, respectively, whereas light blue and blue indicate confidence and high-confidence regions, respectively. **(D)** Indicative alignment of JBP1 in different species. The N-terminus of JBP1 is highly conserved between *Leishmania* species. **(E)** Polarization curves of JBP1 and truncation mutants binding to J-DNA. **(F)** Drawing of the model of the JDBD and THD of JBP1 in complex with J-DNA. The binding sites of base-J and the T base that should be hydroxylated to hmU are located 13 base pairs apart in opposing strands. The grey arrow indicates conformational changes that need to occur to be able to provide a convincing JBP1:J-DNA model. JDBD is colored blue and is based on the JDBD:J-DNA docking model, THD is colored green (C-terminus dark, N-terminus light green) and originated from the AlphaFold model, J-DNA is colored pink and was constructed manually based on the human TET2 homologue structure (PDB-ID: 5DEU).

to DNA. First, we used the THD of the AlphaFold model and the X-ray model of a TET2 homologue in complex with DNA, to create a model of the THD of JBP1 in complex with J-DNA. The THD active site was bound to a thymidine 13 nucleotides downstream base-J, according to the preferred offset between an existing and a new base-J (27). Then, the JDBD:J-DNA complex we obtained above was added to this model by aligning the corresponding DNA regions. However, several modeling attempts did not yield a model that fits with the SAXS data (Fig S14). During the modeling process, it also became apparent that the

N-terminal region in the exact conformation of the AlphaFold model does not allow DNA binding by the JDBD. Based on these findings and as illustrated in Fig 7F, we propose that the combined binding of the THD and JDBD to J-DNA is accompanied by structural rearrangements which likely include DNA bending to allow simultaneous binding of the JDBD and THD-mediated hydroxylation of a thymidine that is located 13 nucleotides downstream in the opposing DNA strand, followed by the N-terminal region wrapping around DNA, and stabilizing the complex.

# Discussion

A computational modeling study helped to identify two new residues in the JDBD of JBP1, Arg448, and Asn455 that are involved in J-DNA binding. After experimental validation by mutational analysis, these residues were added as additional guidelines in docking. This allowed for the construction of a new, improved model for the JDBD:J-DNA complex, that was compatible with SAXS data. Furthermore, the AlphaFold model for full-length JBP1 showed that the individual domains (JDBD and THD) are predicted confidently.

This model allowed us to hypothesize that the N-terminus of JBP1 for many *Trypanosomatidae* species (including Leishmania) is involved in DNA binding, which was confirmed by deletion analysis. The J-DNA-binding residues of the N-terminal region of JBP1 in *L. tarentolae* are likely the conserved positively charged Lys6, Lys7, Lys9, Arg17, ArgR22, and Lys36 (Fig S12). Although these residues are very well conserved in different *Leishmania* and *Leptomonas* species, they are not so in *Trypanosoma* species. This conservation pattern confirms the observation in the PAE matrices of the AlphaFold models (Fig 7A and B) which we discussed above. Furthermore, Asn455, whose mutation to alanine decreases J-DNA binding by more than two orders of magnitude (~400 times) in *L. tarentolae*, is also not conserved in *Trypanosoma* species. Notably, the absence of base-J does not lead to lethality in *Trypanosomes* (6, 35), in sharp contrast to *Leishmania*. It is likely that the lack of the positively charged N-terminal region and Asn455 in *Trypanosomal* JBP1 contribute to lower affinity for J-DNA compared with *Leishmania* JBP1. These, possibly contribute to the difference in lethality that emerges from the loss of base-J in these species. When one considers the possibility that the displacement of RNA polymerase II in TTSs is not mediated by base-J alone, but also by the high affinity of JBP1 for J-DNA in *Leishmania*, it follows that low-affinity *Trypanosomal* JBP1 would not be able to effectively contribute to transcription termination. This might contribute to preventing *Trypanosomes* to evolve dependence for transcriptional termination to the presence of base-J and JBP1.

The AlphaFold model, in combination with the X-ray structure model of TET2 in complex with DNA and our JDBD:J-DNA docking model, still did now allow to construct a reliable model structure for full-length JBP1 bound to J-DNA. Future experiments are required to obtain a structural model for such a complex, which must involve conformational changes between the JDBD and THD domains of JBP1. Such a model would be essential to provide more detailed insights in the molecular mechanism of the unique biochemistry of base-J synthesis that facilitates replication of epigenetic information.

# Materials and Methods

### Macromolecule production and crystallization

Protein expression, crystallization, crystal harvest, and X-ray crystallographic experiment were performed as described before (24). Details are shown in Table 1.

### X-ray crystallography procedures

Crystallographic details are shown in Table 2. The images were processed and scaled at the 17-ID-1 (AMX) beamline of the National Synchrotron Light Source II at Brookhaven (36). The integrated data were scaled in CPP4i2 (37, 38) using AIMLESS (39). The space group P6₁22 was selected and diffraction data extended to 1.95 Å. The structure of JDBD (PDB-ID: 2XSE) was used as search model for molecular replacement. Next, the model was refined in multiple iterative cycles between REFMAC (40) and manual modeling in COOT 0.8.9.2 (41). Within REFMAC, a single TLS body was refined for 15 cycles with starting B-factors fixed to a value of 20.0 Å², jelly body restraints with sigma 0.01, and a maximum distance of 4.2 were set. Typically, 30 cycles were run using riding hydrogens. Optimization of the model based on the PDB-REDO webserver (42) was used to provide additional optimized refinement parameters: simple solvent scaling was used with explicit solvent mask with custom parameters (1.0 increase VDW radius of non-ion atoms, 0.9 increase ionic radius of potential ion atoms, shrink the mask area by 0.9 after calculation). Double conformations were modeled for Ser408, Glu447, Glu488, Met494, and Met546. The final structure was validated using MolProbity (32). The structure has been deposited to the Protein Data Bank (PDB) (43) as entry 8BBM.

### Molecular dynamics simulations of JDBD

#### Protocol

Two independent MD simulations of the JDBD were performed using the GROMACS-2020.2 software (44) and timesteps of 2 fs. The Amber99SB-ILDN force field (45) with periodic boundary conditions was used to describe the system. The topology file was generated from the new JDBD crystal structure (PDB-ID: 8BBM) using the GROMACS *pdb2gmx* tool, after which, the protein was placed in a dodecahedral periodic box and energy minimized with the steepest descent method. Subsequently, 10,341 TIP3 water molecules (46) and 4 Cl⁻ ions were added to solvate the system and neutralize the total charge. Another energy minimization using the steepest descent method was performed. Initial atomic velocities were randomly assigned after Maxwell–Boltzmann distribution (using different random seeds for the two independent simulations). The structure was heated till 300 K in three subsequent *NVT* simulations at 100, 200, and 300 K with position restraint force constants of 10,000, 5,000, and 50 kJ/mol/nm², respectively, to keep the protein backbone Cα atoms positionally restrained with harmonic potentials. Subsequently, an *NpT* simulation was performed (without the use of position restraints) at 1 atm and 300 K for 1 ns, writing coordinates every step, to equilibrate the structure. Next, the MD production run of 100 ns (in which coordinates were written out every five steps) was performed and analyzed. The LINCS algorithm (47) was used to constrain hydrogen-involving bond lengths to their zero-energy value using a single iteration and with the highest order in the expansion of the constraint coupling matrix set to 4. The Langevin integrator was used and set to a friction coefficient for each particle at mass/0.1 ps. The Berendsen barostat (48) was used to keep the pressure close to 1 atm, using a coupling time of 0.5 ps and an isothermal compressibility of 4.5 × 10⁻⁵ bar⁻¹. Short-range electrostatic and van der Waals interactions were evaluated every

**Table 1. JDBD crystallization details.**

| Source organism | *Leishmania tarentolae* |
|---|---|
| Expression host | BL21(DE3)T1$^R$ |
| Crystallization method | Hanging drop |
| Plate type | MRC 2-drop |
| Temperature (K) | 277 |
| Protein concentration | 20 mg/ml |
| Buffer composition of the protein solution | 20 mM HEPES pH 7.5, 140 mM NaCl, 1 mM TCEP |
| Composition of reservoir solution | 15–17% Peg 6000, 0.1 M sodium iodide or 15–17% Peg, 0.2 M potassium nitrate |
| Volume and ratio of drop | 100 nl, 1:1 |
| Volume of reservoir | 150 $\mu$l |

**Table 2. JDBD data collection and processing details.**

| Diffraction source | |
|---|---|
| Wavelength (Å) | 0.98 |
| Space group | P 6$_1$ 2 2 |
| $a, b, c$ (Å) | 68.29, 68.29, 185.84 |
| $\alpha, \beta, \gamma$ (°) | 90.0, 90.0, 120.0 |
| Resolution range (Å) | 59.2–1.95 (2.00–1.95) |
| Total No. of reflections | 753,607 (53,016) |
| No. of unique reflections | 19,605 (1,327) |
| Completeness (%) | 100 (100) |
| Multiplicity | 38.4 (40.0) |
| $\langle I/\sigma(I)\rangle$ | 17.5 (1.0) |
| $R_{r.i.m.}$ | 0.157 (5.033) |
| Overall $B$ factor from Wilson plot (Å$^2$) | 32.3 |
| No. of reflections, working set, test set | 18,522, 1,017 |
| Final $R_{cryst}$, $R_{free}$ | 0.202, 0.252 |
| No. of non-H atoms | |
| Protein | 1,440 |
| Water | 114 |
| R.m.s. deviations/rmsZ Z-score | |
| Bonds | 0.014/0.69 |
| Angles (°) | 1.62/0.76 |
| Average $B$ factors (Å$^2$) | |
| Protein | 55 |
| Water | 54 |
| Ramachandran plot favoured/outliers (%)[a] | 99.4/0.0 |
| Ramachandran Z-score | −0.98 ± 0.72 |
| Rotamers favoured/poor (%)[a] | 96.1/1.3 |
| Clashscore (%-ile)[a] | 100 |
| MolProbity score (%-ile)[a] | 100 |

Values for the outer shell are given in parentheses.
[a]As reported by MolProbity (32).

time step with a distance cut-off of 0.9 nm. The smooth particle mesh Ewald method (49) was used to evaluate long-range electrostatic interactions with a grid spacing of 0.125 nm. Center of mass motion was removed every 10 time steps.

### Analyses

The output trajectories of the MD production runs were processed using the GROMACS-2020.2 *trjconv* tool, keeping every set of coordinates written to disc. Trajectories were visualized using VMD 1.9.3 (50). C$\alpha$ atom positional RMSDs and root-mean-square fluctuations with respect to the corresponding thermally equilibrated structure were calculated over time or averaged over time per residue, respectively, using GROMACS-2020.2 tools *gmx rms* and *gmx rmsf*, respectively. Plots of these time series or values were made using Seaborn 0.11.1 (51). Subsequently, both simulation trajectories were reduced using the GROMACS-2020.2 *trjconv* tool to timesteps of 250 ps. All solvent molecules and ions were removed from these structures to reduce file sizes. The JDBD structures from these reduced trajectories were clustered separately with Bitclust 0.11 (52) utilizing Daura's algorithm (53). The maximum number of clusters to generate was set at 10 clusters per simulation with a RMSD cut-off for pairwise comparisons set at 2.25 Å (52).

### Docking base-J containing DNA to JBP1 with HADDOCK

#### Implementation parameters for base-J

Topology parameters and charges for the base-J residue were generated and implemented in the HADDOCK2.4 webserver (34). For the other DNA nucleotides and the protein, the standard topology parameters and charges defined by default through this webserver were used.

#### Generation of docking input files for base-J-containing DNA

A starting structure for a 20-mer DNA strand containing the sequence 5'-CAGAAGGCAGCJGCAACAAG-3' was created in Pymol using the Build menu (54) in which base-J was built as a regular thymine. Subsequently, this structure was aligned with the coordinates of the structure model published by reference 24 in Coot 0.8.9.2 (41) and the coordinates were saved. Using a text editor, the thymine

residue was replaced by base-J. The resulting file was inspected with YASARA version 20.8.23 (55) and the phosphate backbone bonds with the residues before and after base-J were added. The structure was energy minimized in vacuum followed by energy minimization in explicit TIP3P water molecules, both using the Amber99 force field in YASARA (55).

### Docking protocol

The J-DNA strand was docked to JDBD using the HADDOCK2.4 web server (34). To guide the docking, explicit restraints were added for the distance range between Asp525 OD1 and OD2 with both base-J hydrogen atoms HO4 and HO6 (1.7–3.5 Å). Furthermore, Lys518, Lys522, and Arg532 were manually defined as active residues and Ser515, Arg517, Lys518, Val521, Lys524, Phe528, Lys535 were manually marked as passive residues of JDBD, because mutation studies indicated involvement in formation of the protein:DNA complex for these residues (24). For J-DNA, base-J was manually defined as active residue and the passive residues were defined by the default settings in HADDOCK. The default docking protocol of the HADDOCK2.4 webserver was used and the HADDOCK scoring and clustering was used to rank the output structures (34). J-DNA was docked to several JDBD structures, namely: (i) the new crystal structure; (ii) all central protein structure models of the three most occupied clusters for each MD simulation trajectory that were obtained from clustering; and (iii) the new crystal structure, with Asn455 and Arg448 defined as additional active residues in JDBD. The resulting models were visualized using CCP4 mg (56) and superposed based on the positions of the backbone atoms of J-DNA in such a way that the positions of base-J align.

### Comparison with experimental SAXS and HDX-MS data

For each best-scoring model (cluster1_1.pdb) of the JDBD:J-DNA complex obtained by HADDOCK protocols ii or iii (see the previous section), the SAXS curve was calculated and plotted against the experimentally obtained SAXS curve (24) using CRYSOL from the ATSAS suite (57). In addition, the best-scoring models (cluster1_1.pdb) of the docking models obtained by HADDOCK protocol iii (see the previous section), were compared against the HDX-MS data (24) by mapping the HDX-MS scores onto the protein structures.

## Protein assays

### Protein expression and purification

WT 6xhis-JBP-JDBD protein (*Leishmania Tarentolae*) was expressed in *E. coli* using the pETNKI-his-3C-JBP1-JDBD plasmid and purified as described before (24). Single-point mutations E437A, H440A, R448A, and N455A were introduced in the pETNKI-his-3C-JBP1-JDBD plasmid using a modified QuickChange site-directed mutagenesis method (Agilent). Forward and reverse mutagenesis primers containing the single-point mutations were purchased from Thermo Fisher Scientific: (E437A: 5′-CAT GTG AGC CCA TGC TTT ACC AAC CAG TTC CAG C-3′, H440A: 5′-GGG TTC AGA GCC AGC ATG GCA GCC CAT TCT TTA CCA AC-3′, R448A: 5′-CCA CAG GAA GTC TTT AGC TTC CGG GTT CAG AGC C-3′, N455A: 5′-CAG AGT TCA TTT CAG ACT GGG CTT TCC ACA GGA AGT CTT TAC GTT C-3′). For the mutants, each of the plasmid DNA strands was first amplified in two separate PCR reactions (five

cycles each), one reaction with the forward mutagenesis primer (5′–3′) and a second with the reverse mutagenesis primer. Both reactions were combined and the PCR was continued for another 15 cycles to obtain double-stranded plasmids comprising the single-base substitutions. Plasmids were sequence-verified to confirm the presence of the mutations. Two constructs encoding N-terminal truncations of JBP1 lacking the first 22 amino acid residues (pETNKI-his-3C-JBP123-827, Δ23-JBP1) or the first 37 residues (pETNKI-his-3C-JBP138-827, Δ38-JBP1) were created using ligation-independent cloning (58). Δ23-JBP1 and Δ38-JBP1 proteins were expressed and purified as described for full-length wt-JBP1 (24).

### Fluorescence polarization assays

Fluorescence polarization assays were performed as described before (24). Serial dilutions of wt-JBP1, Δ38-JBP1, Δ23-JBP1, the JDBD, and point-mutated JDBD proteins were prepared on ice in 20 mM Hepes pH 7.5, 140 mM NaCl, 2 mM MgCl$_2$, 1 mM TCEP, 1 mg/ml chicken ovalbumin, 0.05% TWEEN 20, and supplemented with either 1 nM of TAMRA-labeled J-DNA (5′-GGCAGCJGCAACAA-3′) or T-DNA (5′-GGCAGCTGCAACAA-3′). Fluorescence polarization was measured in a PHERAstar FS plate reader (BMG Labtech). Data were analyzed and plotted using Prism (Graphpad).

### Stability of JDBD mutants

The thermal stability of wt-JDBD and mutant proteins was measured in the Prometheus NT.48 nanoDSF (nano differential scanning fluorescence) instrument (NanoTemper Technologies). For each protein, the thermal stability was determined at two concentrations, 0.5 and 0.25 mg ml$^{-1}$ in 20 mM Hepes/HCl at pH 7.5, 140 mm NaCl and 1 mM TCEP. Protein unfolding (ratio of fluorescence at 330 and 350 nm) and aggregation (scattering) was assessed using a temperature slope of 1°C/min in a range from 20°C to 90°C, with an excitation power of 30%. All measurements were performed in duplo.

### Modeling of the JBP1:J-DNA complex using AlphaFold

The AlphaFold2.0 structure of the full-length JBP1 was downloaded (entry Q9U6M1) from the AlphaFold protein structure database (29, 30). To create a model of the THD domain, residues 1–39, and 366–576 (low confidence and JDBD) were removed from the model. Then, the PDB-REDO (42) crystallographic structure of a TET2 homologue of the hydroxylase domain (PDB-ID: 5DEU) was aligned to the THD of this model and the coordinates of loop 113–121 in the AlphaFold model were replaced by the coordinates of the loop 1,291–1,299 of the TET2 homologue. The DNA of the TET2 homologue was used to model THD:J-DNA binding in the Alpha-Fold structure. The 5HC in the DNA was mutated to hydroxymethyluracil (5MU) and the DNA was adjusted to a 25-mer J-DNA with sequence 5′-TCGATTJGTTCATAGACTAATACGT-3′. This THD model in complex with the 25mer J-DNA was energy minimized in vacuum, followed by energy minimization in water using the Amber99 force field (59) and the TIP3P water model in the YASARA software package (55). The ligands oxoglutarate and Fe$^{2+}$ were added to the THD of the newly generated model based on their respective positioning in the TET2 homologous structure model. Next, the JDBD:JDNA model obtained from docking protocol iii (see

the above section) was aligned with the this model in such way that the J-DNA backbone atoms align and the base-J position overlaps. The JDBD was added to the model, and the whole complex was energy minimized in vacuum and water once more. This finally resulted in a JBP1:J-DNA model containing a 25mer J-DNA in complex with the JDBD and THD-JBP1 including the oxoglutarate and $Fe^{2+}$ ligands in the THD.

## Data Availability

Atomic coordinates and structure factors for the reported JDBD crystal structure have been deposited with the Protein Data Bank under accession number 8BBM.

## Supplementary Information

## Acknowledgements

We thank Piet Borst for extensive discussions and suggestions to improve the text of this manuscript, Rosa A Luirink for providing example scripts for the molecular dynamics simulations and its analyses, and Alexander Fish for guidance while performing the fluorescence polarization essays. We thank the NKI's Research High Performance Computing for the computational infrastructure for the molecular dynamics calculations. In addition, we thank Alexandre MJJ Bonvin for implementing the base-J parameters in the HADDOCK2.4 webserver, and Maxim Petoukhov for the advice regarding modeling using CORAL. All students from the Cold Spring Harbor Laboratory class of 2018 are acknowledged for their work on JDBD crystallization and structure modeling. This work was supported by a TOP grant of the Nederlandse Organisatie voor Wetenschappelijk Onderzoek to A Perrakis [grant number 714.014.002] and by an institutional grant of the Dutch Cancer Society and of the Dutch Ministry of Health, Welfare and Sport.

### Author Contributions

I de Vries: conceptualization, formal analysis, validation, investigation, visualization, methodology, and writing—original draft, review, and editing.
D Ammerlaan: investigation.
T Heidebrecht: investigation.
PHN Celie: supervision and investigation.
DP Geerke: supervision and methodology.
RP Joosten: formal analysis, supervision, methodology, and writing—review and editing.
A Perrakis: conceptualization, supervision, funding acquisition, and writing—review and editing.

### Conflict of Interest Statement

The authors declare that they have no conflict of interest.

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
