## [Reviewer comments · Life Science Alliance]

Life Science Alliance

Distant sequence regions of JBP1 contribute to J-DNA binding

Ida de Vries, Danique Ammerlaan, Tatjana Heidebrecht, Patrick Celie, Daan Geerke, Robbie Joosten, and Anastassis Perrakis
DOI: <https://doi.org/10.26508/lsa.202302150>

Corresponding author(s): *Anastassis Perrakis, Netherlands Cancer Institute-Antoni van Leeuwenhoek Hospital*

Review Timeline:

Submission Date:	2023-05-11
Editorial Decision:	2023-05-31
Revision Received:	2023-06-05
Accepted:	2023-06-06

Scientific Editor: *Eric Sawey, PhD*

Transaction Report:

Please note that the manuscript was reviewed at Review Commons and these reports were taken into account in the decision-making process at *Life Science Alliance*.

Review
COMMONS

Review #1

Summary:

The paper presents a combined experimental (X-ray, SAXS, mutational analysis) and computational (MD, docking, AlphaFold) work that elucidates the mechanism of JDBD:JDNA complex formation.

Major comment:

- How did the authors decide the timescale of the production run? Wouldn't the loop motion (which can be necessary for this study) occur on a timescale of 300+ ns?

Minor comments:

- Did I understand correctly that the LINCS algorithm constrained only hydrogen-involving bonds? It is not mentioned explicitly. Or am I missing something?
- The authors should increase the resolution of figures S1 and S3. They look a bit blurry.

The J DNA base is critical for transcription termination at the ends of the polycistronic gene clusters in *Leishmania* and related species. Hence, understanding the formation mechanism of the JDBD:J-DNA complex can provide an opportunity to develop novel chemotherapeutic treatments against these diseases. This work provides the first crystal structure of JDBD with the disordered loop and suggests that R448 and N455, as well as the N-terminus, are involved in the J-DNA binding process. The article is well-written and can interest readers from biological and biochemical societies.

However, my field of expertise is computational chemistry and biochemistry; therefore, I cannot adequately evaluate the accuracy of the experimental techniques used in this work.

Review #2

The manuscript by de Vries et al. reported the crystal structure of the J-DNA binding domain of JBP1. Although, the structure was already solved, the new structure allows to observe a loop that was disordered in the previous structure. This structure was next used to propose models of DNA complex analyzed by MD. The proposed model was then validated by mutagenesis.

Overall, the findings are interesting and the technical quality of the work is high.

Comments:

- For clarity, a figure showing the domain organization of JBP1 could help the reader in the introduction part.
- In addition to Figure 2 showing the newly observed loop in 2Fo-Fc map, an omit map should be included in Supp data.
- Figure S6 legend should be more precise about the type of HDX MS analysis.
- The authors performed MD simulations. But what about DNA curvature upon complex formation?
- p.12 Some mutants were characterized, notably their melting temperatures. One mutant shows decreased stability while R448A shows increased stability. What is the structural explanation?
- Figure 6E: The chi2 value for the comparison of the experimental curve for THD domain and calculated curve is very high indicating a poor fit. What is the explanation?

The novelty of the manuscript mainly relies on the description of the crystal structure of JDBD protein without DNA and proposed models of DNA complex within full length protein, models that were validated by mutations or truncations. The current manuscript well suited for a specialized journal.

Review #3

Base J, also known as β -D-glucopyranosyloxymethyluracil, is a modified form of thymidine that has been identified in kinetoplastids and related organisms. It is worth noting that the distribution of Base J in the genome may vary depending on the organism and its life stage and influences its genome dynamics. The synthesis of this hypermodified nucleotide occurs in two steps, which involve the participation of two distinct thymidine hydroxylases, namely J-binding protein 1 and 2 (JBP1 and JBP2), along with a β -glucosyl transferase. In this study, the authors have presented a crystal structure of JBP1 J-DNA binding domain (J-DBD), which includes a previously reported disordered loop that might be involved in JBP1:J-DNA contact. Using this disordered structure as a starting point, the authors conducted Molecular Dynamics simulations and computational docking studies to propose models for the recognition of J-DNA by JBP1 J-DBD. The results from these studies were validated by punctual mutagenesis experiments, which provided additional data for docking and revealed a binding pattern for JBP1 J-DBD on J-DNA. By combining the crystallographic structure of the TET2 JBP1-homologue in complex with DNA and the AlphaFold model of full-length JBP1, the authors have hypothesized that the flexible N-terminus of JBP1 contributes to DNA-binding, which they have confirmed experimentally. However, according to the authors, to gain a deeper understanding of the unique molecular mechanism that underlies the replication of epigenetic information, an experimental determination of a high-resolution JBP1:J-DNA complex involving conformational changes would be necessary. Nevertheless, the present proposed objectives were fully contemplated by the authors.

****Major comments:****

In my opinion, the present article effectively achieved all the described objectives using appropriate and reproducible methodology, including protein expression and crystallization analysis, Molecular Dynamics analysis using GROMACS-2020.2 software, docking analysis, punctual mutations analyses, and modeling of JBP1:J-DNA complex using the AlphaFold tool. The authors presented the results in a logical and organized manner, making it easy for readers to extract the most important points. However, I believe that the section titled "Results and Discussion" contains more "results" than "discussion". While I understand that the literature on JBPs and base J is still in its early stages, other species of kinetoplastids have JBP1, in which mutations were not lethal as in *L. tarentolae* (e.g. *T. brucei*). Therefore, providing information about the structure of JBP1 and how the present results relate to what is known about JBP1 in other species in terms of structure and J-DNA interactions would significantly enrich the discussion of the findings and reinforce their significance and impact. Thus, the authors should have been clearer about the impact of their findings. When discussing the results, the authors should have answered questions such as how the identification of the new residues involved in JBP1 J-DNA binding impacts the current model of JBP1:J-DNA interactions, how this improved model contributes to the understanding of base J synthesis, and if the new model can be extrapolated to other species of kinetoplastids, according to the conservation of JBP1 among them.

Although the article is more focused on protein research rather than parasite general molecular biology and medical studies, the findings may have implications for the development of new treatments for leishmaniasis. Therefore, the authors should have discussed the potential of their new improved model as a target for lacking treatments of leishmaniasis or at least brought up the point at conclusion section.

****Minor Comments:****

I have some minor comments regarding the text:

1. Please, re-check the affirmation "99% of base-J is found in telomeric repeats, mainly in GGGTTA repeats wherein the second thymine is modified to base-J (2-4)" in the Introduction. I believe that the distribution of base-J varies among different species of trypanosomatids and, therefore, cannot be generalized. Moreover, among different life stages in some organisms such as *T. brucei* and *Leishmania major*, differences on base-J distribution are found. The 99% of telomeric base-J mentioned would be a feature of *Leishmania* genus. Please, re-check the references 3 and 4.
2. Please, enrich the introduction topic with information about the model species, such as importance as pathological agent, its genomic organisation (core, subtelomeres, telomeres, what is present in subtelomeres, including base j) and polycistronic transcription and base J relevance on this aspect. That way, the reader will have a broad and more complete overview of the relevance of the present study.
3. Please, inform the expression vector for *Leishmania tarentolae* JBP1 used to express the mentioned protein on BL21(DE3)T1R.
4. Please, supply the picture of the gel containing the extracted protein.

Overall, this study provides important insights into the JBP1 and DNA interactions, which were lacking in the literature. The use of techniques such as protein expression and crystallization analysis, molecular dynamics, and docking analysis is in line with the research objectives. However, the lack of some information about methodology needs to be addressed (minor comments 3 and 4). Personally, methodology such as molecular dynamics and docking analysis is not easy to critique but the results are clear and understandable.

Although the authors should have been clearer about the impact of their findings, as addressed on my major comments, I believe that protein focused molecular parasitologists would benefit from the finds and methodology presented on this manuscript, since the article is more focused on protein research rather than parasite general molecular biology and medical studies, as mentioned on my major comments.

In summary, this study provides new insights into JBP1 and DNA interactions and uses appropriate and reproducible techniques to achieve its objectives. However, the authors should provide more clarity on the impact of their findings and discuss the potential of their new improved model.

My area of expertise: Cell and molecular biology; stem cells and factor controlling their fate; DNA, RNA, and molecular biology related techniques; Trypanosomatids telomere and telomerase

1. General Statements

We thank the reviewers for the feedback, highlighting the synergy between computational modeling approaches and the experimental techniques we used to study the interaction between JBP1 and J-DNA. We would like to re-iterate that this approach has led to new findings regarding JBP1 and J-DNA interactions, namely:

- (1) We identified and validated an additional interface in DNA binding domain of JBP1 (JDBD), that contributes to J-DNA binding.
- (2) Through analysis of the AlphaFold model of JBP1, we propose how the Thymidine Hydroxylase domain (THD) of JBP1 binds J-DNA, and how the JDBD and THD domains are connected flexibly but explicitly to each other.
- (3) The AlphaFold model allowed to hypothesize that the N-terminus of JBP1 is contributing to J-DNA binding, which we confirmed experimentally.

These findings collectively suggest a mechanistic and structural basis on the synergy between the JDBD, the THD and the N-terminus of JBP1, providing a possible explanation to the previously observed conformational changes of JBP1 upon J-DNA binding. Our findings on the conservation of the N-terminus region and the new interface of JDBD, could be offering an explanation on the differences on how essential base-J is for different *Trypanosomidae* species. They also offer a first glimpse of how these domains synergize to provide new insights in the semi-conservative replication mechanisms of the base-J epigenetic marker in kinetoplastids.

Reviewer #1: Evidence, reproducibility and clarity

Summary:

The paper presents a combined experimental (X-ray, SAXS, mutational analysis) and computational (MD, docking, AlphaFold) work that elucidates the mechanism of JDBD:JDNA complex formation.

Major comment:

- How did the authors decide the timescale of the production run? Wouldn't the loop motion (which can be necessary for this study) occur on a timescale of 300+ ns?

While our simulations showed overall stability of the simulated protein (as reflected by the RMSD time series), the RMSF provided clear indications for differences in flexibility in the loops and termini of JBP1. We believe that performing an MD simulation of 100 ns *in duplo* samples the flexibility and behavior of the JBP1 DNA binding domain (JDBD) sufficiently for obtaining templates for docking studies, which was the purpose of running the simulations. We emphasized this in the manuscript (page 10) by adding: *"and obtain additional templates for further docking studies"*.

Minor comments:

- Did I understand correctly that the LINCS algorithm constrained only hydrogen-involving bonds? It is not mentioned explicitly. Or am I missing something?

LINCS was indeed used to only constrain hydrogen-involving bonds. We made this more explicit in the MD protocol described in the method section of the manuscript: *"The LINCS algorithm (40) was used to constrain hydrogen-involving bond lengths to their zero-energy value"*.

- The authors should increase the resolution of figures S1 and S3. They look a bit blurry.

We apologize for that; we tried to improve that by adjusting the figure sizes, but we are constrained by the output from e.g., the Bitclust software. We sincerely hope that the current resolution does not present an obstacle to the reader.

Significance:

The J DNA base is critical for transcription termination at the ends of the polycistronic gene clusters in *Leishmania* and related species. Hence, understanding the formation mechanism of the JDBD:J-DNA complex can provide an opportunity to develop novel chemotherapeutic treatments against these diseases. This work provides the first crystal structure of JDBD with the disordered loop and suggests that R448 and N455, as well as the N-terminus, are involved in the J-DNA binding process. The article is well-written and can interest readers from biological and biochemical societies. However, my field of expertise is computational chemistry and biochemistry; therefore, I cannot adequately evaluate the accuracy of the experimental techniques used in this work.

We thank the reviewer, but would like to emphasize that our work goes well beyond the new interface of the JDBD, offering significant new insights on the synergy between the THD, the JDBD and the newly identified N-terminus binding to J-DNA, as also outlined in the general summary above.

Reviewer #2: Evidence, reproducibility and clarity

The manuscript by de Vries et al. reported the crystal structure of the J-DNA binding domain of JBP1. Although, the structure was already solved, the new structure allows to observe a loop that was disordered in the previous structure. This structure was next used to propose models of DNA complex analyzed by MD. The proposed model was then validated by mutagenesis.

Overall, the findings are interesting and the technical quality of the work is high.

We believe, as we outlined in the summary, that our work goes well beyond showing the new structure and validate the new interface by mutagenesis. In our view, the major findings of the paper have to do with the AlphaFold modeling analysis and validation, and the finding that the N-terminus of JBP1 is involved in DNA binding, something that is not only new, but has also been totally unexpected.

Comments:

-For clarity, a figure showing the domain organization of JBP1 could help the reader in the introduction part.

The domain architecture of JBP1 was added to Figure 1 as panel B.

-In addition to Figure 2 showing the newly observed loop in 2Fo-Fc map, an omit map should be included in Supp data.

A figure of the omit map was added to the supplemental information as Supplemental Figure S1.

-Figure S6 legend should be more precise about the type of HDX MS analysis.

As the HDX-MS data and methods were described in detail in previous work (Heidebrecht *et al.* 2011), we left the details out in the current manuscript. The reference to this paper was added to Figure S6 for clarity.

-The authors performed MD simulations. But what about DNA curvature upon complex formation?

For the JDBD MD simulations, we did not add at all the (J-)DNA and the current simulations provide no information about its curvature. As mentioned in the discussion, we do expect conformational changes when the J-DNA:JBP1 complex forms, and this likely includes DNA curvature as well as conformational changes between the protein domains. We felt that the current data would not allow to extract new insights from such complicated simulations.

-p.12 Some mutants were characterized, notably their melting temperatures. One mutant shows decreased stability while R448A shows increased stability. What is the structural explanation?

Indeed, the E437A mutant (that showed lower expression compared to the other mutants) showed decreased thermal stability. The R448A shows an increase in stability ($\sim 3^\circ$ C), and so does the H440 mutant ($\sim 2^\circ$ C). While there is no specific structural explanation for these observations, in general mutation to alanine reduces the entropy-loss upon protein folding. The reason we comment about the stability is to point out that the dramatically decreased binding of the N455A and R448A mutants is not due to a decrease of the protein stability. This is now clarified in the manuscript: *“the other mutants are as stable or slightly more stable compared to the wild-type, suggesting that the DNA-binding analysis is not affected significantly by altered protein stability.”*

-Figure 6E: The chi² value for the comparison of the experimental curve for THD domain and calculated curve is very high indicating a poor fit. What is the explanation?

The χ^2 value indeed reflects differences between the JBP1 THD selected from the AlphaFold model and the structure used in the SAXS experiment. The experimental model is the so-called Δ DBD, which is the full length JBP1, where the JDBD is missing (it has been “spliced out”). Hence, the connecting loops and the N-terminus are present in this Δ DBD structure, whereas in the THD of the AlphaFold model, these parts of the structure were left out. In other words, while the full-length computational model refers to the exact same purified protein, the computationally truncated model and the purified protein for the experiments, have actual differences. Thus, the shape and fit of the experimental curve to the calculated curve can be considered pretty good. This is now clarified in the manuscript by adding: *“The χ^2 value of the fit is slightly elevated due to presence of the connecting loops between the THD and the JDBD and the N-terminus in the protein used for measuring the SAXS curve, which were removed from the computational model.”*

Significance:

The novelty of the manuscript mainly relies on the description of the crystal structure of JDBD protein without DNA and proposed models of DNA complex within full length protein, models that were validated by mutations or truncations. The current manuscript well suited for a specialized journal.

These findings are indeed novel, especially the discovery and validation of the new interface of JDBD to J-DNA. We want to iterate that we are most excited by the totally unexpected and mechanistically important discovery of the role of the N-terminus of JBP1, that brings together legacy data and raises interest for additional structural studies.

Reviewer #3: Evidence, reproducibility and clarity

Base J, also known as β -D-glucopyranosyloxymethyluracil, is a modified form of thymidine that has been identified in kinetoplastids and related organisms. It is worth noting that the distribution of Base J in the genome may vary depending on the organism and its life stage and influences its genome dynamics. The synthesis of this hypermodified nucleotide occurs in two steps, which involve the participation of two distinct thymidine hydroxylases, namely J-binding protein 1 and 2 (JBP1 and JBP2), along with a β -glucosyl transferase. In this study, the authors have presented a crystal structure of JBP1 J-DNA binding domain (J-DBD), which includes a previously reported disordered loop that might be involved in JBP1:J-DNA contact. Using this disordered structure as a starting point, the authors conducted Molecular Dynamics simulations and computational docking studies to propose models for the recognition of J-DNA by JBP1 J-DBD. The results from these studies were validated by punctual mutagenesis experiments, which provided additional data for docking and revealed a binding pattern for JBP1 J-DBD on J-DNA. By combining the crystallographic structure of the TET2 JBP1-homologue in complex with DNA and the AlphaFold model of full-length JBP1, the authors have hypothesized that the flexible N-terminus of JBP1 contributes to DNA-binding, which they have confirmed experimentally. However, according to the authors, to gain a deeper understanding of the unique molecular mechanism that underlies the replication of epigenetic information, an experimental determination of a high-resolution JBP1:J-DNA complex involving conformational changes would be necessary. Nevertheless, the present proposed objectives were fully contemplated by the authors.

Major comments:

In my opinion, the present article effectively achieved all the described objectives using appropriate and reproducible methodology, including protein expression and crystallization analysis, Molecular Dynamics analysis using GROMACS-2020.2 software, docking analysis, punctual mutations analyses, and modeling of JBP1:J-DNA complex using the AlphaFold tool. The authors presented the results in a logical and organized manner, making it easy for readers to extract the most important points. However, I believe that the section titled "Results and Discussion" contains more "results" than "discussion". While I understand that the literature on JBPs and base J is still in its early stages, other species of kinetoplastids have JBP1, in which mutations were not lethal as in *L. tarentolae* (e.g. *T. brucei*). Therefore, providing information about the structure of JBP1 and how the present results relate to what is known about JBP1 in other species in terms of structure and J-DNA interactions would significantly enrich the discussion of the findings and reinforce their significance and impact. Thus, the authors should have been clearer about the impact of their findings. When discussing the results, the authors should have answered questions such as how the identification of the new residues involved in JBP1 J-DNA binding impacts the current model of JBP1:J-DNA interactions, how this improved model contributes to the understanding of base J synthesis, and if the new model can be extrapolated to other species of kinetoplastids, according to the conservation of JBP1 among them.

Although the article is more focused on protein research rather than parasite general molecular biology and medical studies, the findings may have implications for the development of new treatments for leishmaniases. Therefore, the authors should have discussed the potential of their new improved model as a target for lacking treatments of leishmaniases or at least brought up the point at conclusion section.

We thank the reviewer for pointing out that the comparison between kinetoplastid species could be described more explicit to highlight the impact of the presented results with respect to the

variety in JBP1 sequence, and possibly contribute to understanding the role of base-J in the differences in lethality and transcription regulation within these species. We now elaborate on our results in multiple places in the manuscript:

- In the introduction we introduce the differences in lethality and transcription regulation within *Leishmania* and *Trypanosoma* (see also minor comment 2 below).
- An alignment of full-length JBP1 sequences of different *Trypanosomatidae* species was added as Supplemental Figure S10 to support the discussion below.
- The section describing the docking model of JDBD:J-DNA has been amended with a description regarding the conservation of the residues involved in the binding interface: *"The residues described in the JDBD:J-DNA interface are in general highly conserved (Supplemental Figure S12). Asp525 is fully conserved in Leishmania, Trypanosoma, Leptomonas and Bodo saltans species, so are Lys522, Arg532 and ArgR448. Asn455, which we identify in this study, is not conserved in Leptomonas, and Arg517 is not conserved in Trypanosoma also."*
- We renamed the final section to "Conclusions and Outlook" and added some discussion focusing on the conservation of the residues in the N-terminus and in the JDBD between different *Trypanosomatidae* species. Specifically, we now discuss how these could contribute in understanding the differences in lethality and transcription termination between *Leishmania* and *Trypanosoma* in the absence of base-J.

Minor Comments:

1. Please, re-check the affirmation "99% of base-J is found in telomeric repeats, mainly in GGGTTA repeats wherein the second thymine is modified to base-J (2-4)" in the Introduction. I believe that the distribution of base-J varies among different species of trypanosomatids and, therefore, cannot be generalized. Moreover, among different life stages in some organisms such as *T. brucei* and *Leishmania major*, differences on base-J distribution are found. The 99% of telomeric base-J mentioned would be a feature of *Leishmania* genus. Please, re-check the references 3 and 4.

Indeed, the referee is right to mention that the 99% is a *Leishmania*-specific observation. We have modified the introduction accordingly. *"Base-J is found mainly in telomeric repeats and other repetitive sequences. In Leishmania 99% of base-J is found in telomers, mainly in GGGTTA repeats, wherein the second thymine is modified to base-J (2–4)."*

2. Please, enrich the introduction topic with information about the model species, such as importance as pathological agent, its genomic organisation (core, subtelomeres, telomeres, what is present in subtelomeres, including base j) and polycistronic transcription and base J relevance on this aspect. That way, the reader will have a broad and more complete overview of the relevance of the present study.

We have enriched the introduction with a paragraph (*"Leishmania species are uni-cellular [...] essentiality of base-J for the life circle of these parasites."*) outlining the issues raised by the referee. As suggested by the referee, this makes it easier to both understand the relevance of the present study and to enrich the discussion about our findings discussed earlier in this letter.

3. Please, inform the expression vector for *Leishmania tarentolae* JBP1 used to express the mentioned protein on BL21(DE3)T1R.

The expression vector for JDBD JBP1 used for the crystallization was mentioned in Heidebrecht *et al.* 2011, which we refer to in the text. For clarity we added the vector to the first sentence in

Full Revision

the protein expression and purification section in the material and methods: “using the pETNKHIS-3C-JBP1-JDBD plasmid”.

4. Please, supply the picture of the gel containing the extracted protein.

The gel of the JDBD mutants and the JBP1 N-terminus truncations was added to the supplemental information as Figure S10.

Significance:

Overall, this study provides important insights into the JBP1 and DNA interactions, which were lacking in the literature. The use of techniques such as protein expression and crystallization analysis, molecular dynamics, and docking analysis is in line with the research objectives. However, the lack of some information about methodology needs to be addressed (minor comments 3 and 4). Personally, methodology such as molecular dynamics and docking analysis is not easy to critique but the results are clear and understandable.

Although the authors should have been clearer about the impact of their findings, as addressed on my major comments, I believe that protein focused molecular parasitologists would benefit from the finds and methodology presented on this manuscript, since the article is more focused on protein research rather than parasite general molecular biology and medical studies, as mentioned on my major comments.

In summary, this study provides new insights into JBP1 and DNA interactions and uses appropriate and reproducible techniques to achieve its objectives. However, the authors should provide more clarity on the impact of their findings and discuss the potential of their new improved model.

My area of expertise: Cell and molecular biology; stem cells and factor controlling their fate; DNA, RNA, and molecular biology related techniques; Trypanosomatids telomere and telomerase

We would like to thank the referee for his positive and constructive outlook, which we believe resulted in changes that put the impact of our findings in clearer perspective.

May 31, 2023

RE: Life Science Alliance Manuscript #LSA-2023-02150

Anastassis Perrakis
The Netherlands Cancer Institute
Biochemistry (B8)
Plesmanlaan 121
Amsterdam 1066 CX
Netherlands

Dear Dr. Perrakis,

Thank you for submitting your revised manuscript entitled "Distant sequence regions of JBP1 contribute to J-DNA binding". We would be happy to publish your paper in Life Science Alliance pending final revisions necessary to meet our formatting guidelines.

- please upload your main manuscript text as an editable doc file
- Please upload all figure files as individual ones, including the supplementary figure files; all figure legends should only appear in the main manuscript file
- please add your main, supplementary figure, and table legends to the main manuscript text after the references section
- please add a Running Title and a Summary Blurb/Alternate Abstract to our system
- please add a Category for your manuscript in our system
- please add the Twitter handle of your host institute/organization as well as your own or/and one of the authors in our system
- please consult our manuscript preparation guidelines <https://www.life-science-alliance.org/manuscript-prep> and make sure your manuscript sections are in the correct order
- please upload your Tables in editable .doc or excel format. They can be included at the bottom of the main manuscript file or sent as separate files.
- conclusion section should be provided as a part of the Results and Discussions section and not entered separately. Alternatively, you can rename Results and Discussion as just Results, and rename the Conclusions and Outlook section as Discussion
- please add an Author Contributions section to your main manuscript text and in the system
- in the manuscript text, there is a callout for figure S2B even though the figure doesn't have any panels. Kindly correct this.
- please add callouts for Figures S4A-D, S6A-F, S7A-F and Table S1 to your main manuscript text
- the **ACCESSION NUMBERS** section should be renamed Data Availability Statement, and placed at the end of the Materials & Methods section

A. FINAL FILES:

B. MANUSCRIPT ORGANIZATION AND FORMATTING:

Sincerely,

Reviewer #1 (Comments to the Authors (Required)):

I have no further questions to the authors.

Reviewer #2 (Comments to the Authors (Required)):

Base J, also known as β -D-glucopyranosylmethyluracil, is a modified form of thymidine that has been identified in kinetoplasts and related organisms. It is worth noting that the distribution of Base J in the genome may vary depending on the organism and its life stage and influence its genome dynamics. The synthesis of this hypermodified nucleotide occurs in two steps, which involve the participation of two distinct thymidine hydroxylases, namely J-binding protein 1 and 2 (JBP1 and JBP2), along with a β -glucosyl transferase. In this study, the authors have presented a crystal structure of JBP1 J-DNA binding domain (J-DBD), which includes a previously reported disordered loop that might be involved in JBP1:J-DNA contact. Using this disordered structure as a starting point, the authors conducted Molecular Dynamics simulations and computational docking studies to propose models for the recognition of J-DNA by JBP1 J-DBD. The results from these studies were validated by punctual mutagenesis experiments, which provided additional data for docking and revealed a binding pattern for JBP1 J-DBD on J-DNA. By combining the crystallographic structure of the TET2 JBP1-homologue in complex with DNA and the AlphaFold model of full-length JBP1, the authors have hypothesized that the flexible N-terminus of JBP1 contributes to DNA-binding, which they have confirmed experimentally. However, according to the authors, to gain a deeper understanding of the unique molecular mechanism that underlies the replication of epigenetic information, an experimental determination of a high-resolution JBP1:J-DNA complex involving conformational changes would be necessary. Nevertheless, the present proposed objectives were fully contemplated by the authors. Moreover, the authors successfully improved the manuscript according to the point by point suggestions of the three reviewers, turning the manuscript now suitable for publication in the current format.

June 6, 2023

RE: Life Science Alliance Manuscript #LSA-2023-02150R

Prof. Anastassis Perrakis
Netherlands Cancer Institute-Antoni van Leeuwenhoek Hospital
Biochemistry (B8)
Plesmanlaan 121
Amsterdam 1066 CX
Netherlands

Dear Dr. Perrakis,

Thank you for submitting your Research Article entitled "Distant sequence regions of JBP1 contribute to J-DNA binding". It is a pleasure to let you know that your manuscript is now accepted for publication in Life Science Alliance. Congratulations on this interesting work.

DISTRIBUTION OF MATERIALS:

Again, congratulations on a very nice paper. I hope you found the review process to be constructive and are pleased with how the manuscript was handled editorially. We look forward to future exciting submissions from your lab.

Sincerely,
